# Effect of Non-Modified as Well as Surface-Modified SiO_2_ Nanoparticles on Red Blood Cells, Biological and Model Membranes

**DOI:** 10.3390/ijms241411760

**Published:** 2023-07-21

**Authors:** Katarzyna Solarska-Ściuk, Katarzyna Męczarska, Vera Jencova, Patryk Jędrzejczak, Łukasz Klapiszewski, Aleksandra Jaworska, Monika Hryć, Dorota Bonarska-Kujawa

**Affiliations:** 1Department of Physics and Biophysics, Wrocław University of Environmental and Life Sciences, Norwida St. 25, 50-375 Wrocław, Poland; katarzyna.meczarska@upwr.edu.pl (K.M.); 107268@student.upwr.edu.pl (A.J.); 107264@student.upwr.edu.pl (M.H.); dorota.bonarska-kujawa@upwr.edu.pl (D.B.-K.); 2Department of Chemistry, Faculty of Science, Humanities and Education, Technical University of Liberec, Studentska 2, 461 17 Liberec, Czech Republic; vera.jencova@tul.cz; 3Institute of Chemical Technology and Engineering, Faculty of Chemical Technology, Poznan University of Technology, Berdychowo 4, 60-965 Poznan, Poland; 1patryk.jedrzejczak@gmail.com (P.J.); lukasz.klapiszewski@put.poznan.pl (Ł.K.)

**Keywords:** silica nanoparticles, hemolytic toxicity, osmotic resistance, biological membranes, erythrocytes, erythrocyte membranes, liposomes

## Abstract

Nanoparticles are extremely promising components that are used in diagnostics and medical therapies. Among them, silica nanoparticles are ultrafine materials that, due to their unique physicochemical properties, have already been used in biomedicine, for instance, in cancer therapy. The aim of this study was to investigate the cytotoxicity of three types of nanoparticles (${\mathrm{SiO}}_{2}$, ${\mathrm{SiO}}_{2}\text{-}\mathrm{SH}$, and ${\mathrm{SiO}}_{2}\text{-}\mathrm{COOH}$) in relation to red blood cells, as well as the impact of silicon dioxide nanoparticles on biological membranes and liposome models of membranes. The results obtained prove that hemolytic toxicity depends on the concentration of nanoparticles and the incubation period. Silica nanoparticles have a marginal impact on the changes in the osmotic resistance of erythrocytes, except for ${\mathrm{SiO}}_{2}\text{-}\mathrm{COOH}$, which, similarly to ${\mathrm{SiO}}_{2}$ and ${\mathrm{SiO}}_{2}\text{-}\mathrm{SH}$, changes the shape of erythrocytes from discocytes mainly towards echinocytes. What is more, nanosilica has an impact on the change in fluidity of biological and model membranes. The research gives a new view of the practical possibilities for the use of large-grain nanoparticles in biomedicine.

## 1. Introduction

Recently, ${\mathrm{SiO}}_{2}$ nanoparticles have become easily accessible, which makes them common in numerous products. Furthermore, they are used in the chemical industry as an ingredient in cosmetics, drugs, printer toner, varnishes, and food. Silicon dioxide nanoparticles are also more and more popular in both therapeutic and diagnostic applications of medicine [1,2]. Biomedical and biotechnological studies have focused on the use of silicon dioxide nanoparticles in therapies, and diagnostics has experienced much progress, including in cancer therapies, drug delivery, and enzyme deactivation [3]. The majority of mesoporous silica nanoparticles (MSN) have pores with a size of about 2–50 nm [4] and are described as well-organized structures, which makes precise drug delivery possible and allows one to control the kinetics of drug release. The average surface of the pores is more than 700 $m^{2}$/g, which exceeds the outer size of non-pore nanoparticles, and there is a strong attraction between the pores of nanoparticles. It is also possible to modify the surface of MSN, and, consequently, their properties, as a way of introducing additional functional groups [5]. As a result of that process, silica nanoparticles are able to enter into reactions with ions, atoms, and other particles using their whole volume [6]. One advantage of using mesoporous silica nanoparticles is their excellent biocompatibility, which explains why silicon has been viewed by the United States Food and Drug Administration (FDA) as a generally safe element. There are two functional regions of these nanoparticles, namely, pores (which are well organized) and the outer surface, whose numerous functional groups give them a high level of reactivity. As a result of surface modification with functional groups, it is possible to obtain nanoparticles that show desired properties in order to obtain better control in the process of drug delivery. As a result of the significant volume of pores, it is also possible to use them as drug carriers [5,6]. In order to ensure both the safety and biocompatibility of selected nanoparticles, it is necessary to research their impact on the morphotic components of blood with which they are in direct contact due to the fact that a drug is delivered in an intravenous manner [7]. The components mentioned above include erythrocytes and cell membranes. Erythrocytes are responsible for delivering oxygen to cells, as well as removing carbon dioxide to the lungs [8]. They have a dyscocytic shape, which may change due to numerous physical and chemical factors; therefore, they are able to enter into interactions depending on the physiological environment in which they are located. There are two forms of them, namely, physiological and pathological. The reversible transformation of dyscocytes in stomatocytes (cells that are concave on one side) is observed as a result of a reaction with amphipathic substances that accumulate in the inner monolayer of the lipid bilayer in a cell, under the influence of acid pH and as a result of high hydrostatic pressure. In contrast, reversible transformation of dyscocytes in echinocytes (round cells with numerous pores) is possible thanks to amphipathic substances that accumulate in the outer monolayer of the lipid bilayer of a cell, the lower level of ATP, an excessive amount of cholesterol, and the influence of alkaline pH [9]. From a structural viewpoint, the most important component of an erythrocyte is the cell membrane, which is responsible for proper antigenic transport, and the mechanical properties of the entire cell. Its structure is not significantly different from the structure of the other cell membranes and cell organelles as, similarly to them, it consists of phospholipids and proteins. The types of lipids present in the cell membrane of erythrocytes are mainly cholesterol, phospholipids, and glycolipids [10]. As a result of both the structure and chemical composition of biological membranes being diversified (which makes it complicated to conduct studies or structural and physicochemical analyses), it is vital to use only small bubbles that consist of phospholipids, which are called liposomes [11,12,13]. In scientific research, these bubbles are a model of biological membranes [11,12,14]. To the best of our knowledge, there have been no comprehensive studies or analyses yet of the important interaction between mesoporous silica nanoparticles and red blood cells (RBCs), or their direct effect on both biological membranes and liposome models of membranes. Therefore, the aims of the experiment were to determine the hemolytic activity [15,16] of three selected silicon nanoparticles regarding red blood cells, and to study the impact they have on the physical parameters of erythrocyte membranes and liposomes. It was also essential to check the effect of nanoparticles on the integrity of the erythrocyte membrane by monitoring whether the nanoparticles deposited on the erythrocyte could cause damage to their membrane. In the study both unmodified mesoporous ${\mathrm{SiO}}_{2}$ particles and those that had been modified on their surface using functional groups (-SH (${\mathrm{SiO}}_{2}\text{-}\mathrm{SH}$) and -COOH (${\mathrm{SiO}}_{2}\text{-}\mathrm{COOH}$)) were selected. Silica nanoparticles have the ability to adsorb onto the surface of red blood cells (RBCs) through various forces, including electrostatic interactions and hydrogen bonding. This adsorption occurs due to the hydrophilic nature of the silica surface and the presence of hydrophilic groups on the RBC membrane. In certain cases, smaller silica nanoparticles with appropriate surface properties can even penetrate the RBC membrane. This can take place when the nanoparticles are small enough to fit through the pores or defects in the cell membrane, or if there are specific interactions between the nanoparticle surface and the membrane components. The interaction between silica nanoparticles and RBCs can be influenced by the functional groups present on the surface of the nanoparticles. For instance, if the surface is modified with positively charged groups, they may interact with negatively charged components on the RBC membrane, resulting in specific binding or altered membrane properties. Moreover, depending on the nature and concentration of the silica nanoparticles, they may induce hemolysis, which refers to the rupture of RBCs. This hemolytic effect can occur when the nanoparticles disrupt the integrity of the cell membrane or induce osmotic imbalances (Figure 1). Another crucial consideration is that the interaction of silica nanoparticles with RBCs can trigger cellular responses. This includes the activation of signaling pathways, release of inflammatory mediators, or oxidative stress, which can influence the overall biocompatibility of the nanoparticles. It is important to emphasize that the specific details of this interaction can vary based on a wide range of factors, such as the size, shape, surface charge, and surface chemistry of the silica nanoparticles, as well as the physiological conditions in which the interaction takes place [17,18].

The size of grains in the selected nanoparticles was not larger than 200 nm, and the diameter of pores was constant and equaled 4 nm. They were also more carefully studied from the point of view of physical and chemical methods using ATR-FTIR to determine the quality of functional groups that are present in the structures of pure silica, which was functionalized using propylcarboxylic acid and propylthiol. The particle size and electrokinetic potential were analyzed using a Zetasizer Nano ZS. The studies presented in this manuscript proved that surface functionalization caused larger particles than nonfunctionalized silica. The main reason for this is the possibility of these nanomaterials joining into aggregates and, consequently, agglomerates. What is more, all the tested samples showed a negative electrokinetic value. Furthermore, the obtained results showed that both unmodified and modified ${\mathrm{SiO}}_{2}$ nanoparticles have hemolytic toxicity that depends on the period of incubation and concentration. Moreover, they induce changes in the shapes of erythrocytes, mainly causing the formation of echinocytes. It was also proven that there is a link between the changes in the structure of a membrane and the concentration of the particles used—namely, even the slightest changes in the fluidity of cell membranes are responsible for changes in their function, including permeability. The experiments conducted in this paper prove that the ${\mathrm{SiO}}_{2}$ nanoparticles used are an interesting material in the field of nanomedicine and biomaterials due to their morphology, simple functionalization of the surface structure, porosity, stability, biocompatibility, biodegradability, and the fact that they can be used in theranostics. Despite all the advantages mentioned, their hemolytic effect must be considered and depends on the type of nanoparticle, dosage, and incubation time.

## 2. Results

### 2.1. Attenuated Total Reflectance Infrared Spectroscopy Technique (ATR-FTIR)

As a result of the ATR-FTIR analysis, it was possible to determine the quality of the functional groups present in the analyzed samples. The results obtained are presented in Figure 2 in the form of infrared spectra.

While analyzing the spectra obtained from all of the samples, it was observed that the most intensive wavebands appeared in all the samples tested. The high-intensity band at 1060 ${\mathrm{cm}}^{-1}$ and the low-intensity band at 960 ${\mathrm{cm}}^{-1}$ were both due to vibrations of the stretching formations of Si-O and O-Si-O, a phenomenon that is typical for silica [17,18,19,20,21,22]. Two more bands were present in the spectrum of the tested silica at the wavenumbers 798 ${\mathrm{cm}}^{-1}$ and 452 ${\mathrm{cm}}^{-1}$, and they were connected with the stretching vibrations of Si-O-Si groups and the bending vibrations of Si-O groups of siloxane groups, respectively [22,23,24]. Finally, in regard to functionalized silica, an additional band at the wavenumber 2979 ${\mathrm{cm}}^{-1}$ was observed, which was probably due to the vibrations of C-H stretching groups that were present in the aliphatic chains connected to the surface of the tested silica dioxides [25,26].

### 2.2. Particle Size Ranges Measurement

In order to determine the dispersive properties of the analyzed silicon dioxides, the particle size distributions were measured (Figure 3). As shown in the attached charts, functionalization using both propylcarboxylic acid and propylthiol was responsible for the creation of particles with a larger size than nonfunctionalized silica. This may be connected to the stronger tendency of these materials to join into aggregates and, consequently, agglomerates [27]. With regard to the sample of pure silica, the size of particles was within the 142–342 nm range, and the greatest number of particles, namely 30.1%, were 220 nm. The size range of the silica nanoparticles that were functionalized using propylcarboxylic acid was 255–712 nm, and for silica functionalized using propylthiol it was 396–1006 nm. With regard to ${\mathrm{SiO}}_{2}\text{-}\mathrm{COOH}$, the most frequently observed size of particles was 459 nm (25.8% of the total). With regard to ${\mathrm{SiO}}_{2}\text{-}\mathrm{SH}$, the predominant size was 615 nm (29.3% of the total).

### 2.3. Electrokinetic Potential Measurement

In the next step, electrokinetic potential measurement of the tested silica samples, namely pure silica and silica functionalized using propylcarboxylic acid and propylthiol, was conducted. The results obtained are presented in Figure 4. Although all the tested samples of silicon dioxides in a buffer solution with a pH of 7.33 showed negative electrokinetic values, in the functionalized samples, this value was more negative. For the tested samples of ${\mathrm{SiO}}_{2}$, ${\mathrm{SiO}}_{2}\text{-}\mathrm{COOH}$, and ${\mathrm{SiO}}_{2}\text{-}\mathrm{SH}$, the electrokinetic potential was −28.6 mV, −34.6 mV, and −30.2 mV, respectively. On the basis of these values, it is predicted that, although the tested silica samples, under a given pH, create a stable electrokinetic dispersion, the best seems to be the ${\mathrm{SiO}}_{2}\text{-}\mathrm{COOH}$ sample. The zeta potential for the tested silica samples was in line with the literature data [28,29]. In addition, as proven in the literature, the value of the ${\mathrm{SiO}}_{2}$ electrokinetic potential depends on the method of silica synthesis, the conditions of storage, and the surface properties.

### 2.4. Hemolysis Assay of Erythrocytes

In the first step, the aim was to determine the cytotoxic activity of ${\mathrm{SiO}}_{2}$, ${\mathrm{SiO}}_{2}\text{-}\mathrm{SH}$, and ${\mathrm{SiO}}_{2}\text{-}\mathrm{COOH}$ nanoparticles in relation to red blood cells. In control (untreated) samples, it was observed that hemolysis reached 3.09% after 2 h of incubation and 3.79% after 24 h of incubation (Figure 5). Moreover, when analyzing the range concentrations used (0–1000 μg/mL), the unmodified ${\mathrm{SiO}}_{2}$ nanoparticles had a tendency to show hemolytic toxicity and a correlation between hemolysis and the highest concentration (21.39% after 2 h of incubation, and 97.09% after 24 h of incubation). The data showed that a large number of red blood cells (RBCs) were hemolyzed. More generally, this study proved that all of the tested silica nanoparticles generate hemolysis, which is concentration- and time-dependent. In another case, when the concentration of ${\mathrm{SiO}}_{2}\text{-}\mathrm{SH}$ nanoparticles increased, the observed level of hemolysis similarly increased (proven after 2 and 24 h of interaction between nanoparticles and red blood cells). Furthermore, for all the tested silica nanoparticles incubated for 24 h, the level of hemolysis was significantly higher than after 2 h of incubation for the same concentrations. Further interesting results were obtained for ${\mathrm{SiO}}_{2}\text{-}\mathrm{COOH}$ nanoparticles, for which the level of hemolysis after 2 h of incubation was 6.04% and, after 24 h of incubation, 6.88%. Compared to the data for ${\mathrm{SiO}}_{2}$ and ${\mathrm{SiO}}_{2}\text{-}\mathrm{SH}$, these (${\mathrm{SiO}}_{2}\text{-}\mathrm{COOH}$) silica nanoparticles are not only safer for erythrocytes, but also nontoxic at the highest concentrations.

### 2.5. Osmotic Resistance of Erythrocytes

In the experiment that followed, the osmotic resistance of untreated RBCs (control cells), u-modified silica nanoparticles (${\mathrm{SiO}}_{2}$), and surface-modified silica nanoparticles (${\mathrm{SiO}}_{2}\text{-}\mathrm{SH}$ and ${\mathrm{SiO}}_{2}\text{-}\mathrm{COOH}$) was studied. Compared to the previous study, which concentrated more on the hemolytic effect of nanoparticles (for the range of concentrations from 0 to 1000 μg/mL), in this experiment, we used two nonlytic concentrations, namely 200 μg/mL and 500 μg/mL. Figure 6 illustrates the curves of osmotic resistance recorded in erythrocytes modified using ${\mathrm{SiO}}_{2}$ nanoparticles. It is possible to determine the level of osmotic resistance in relation to the concentration of psychological saline (${\mathrm{IC}}_{50}$), which is responsible for the 50% hemolysis of these RBCs, which were modified by ${\mathrm{SiO}}_{2}$ nanoparticles (Table 1). It has been proven experimentally that 50% hemolysis of erythrocytes not treated with silica nanoparticles was observed at a NaCl concentration of 0.69% (after 2 h of incubation) and 0.66% after 24 h of incubation. In contrast, 50% hemolysis of red blood cells incubated with ${\mathrm{SiO}}_{2}$ (500 μg/mL) nanoparticles was noticed at a NaCl concentration of 0.71% after 2 h of incubation and 0.73% after 24 h of incubation. For ${\mathrm{SiO}}_{2}\text{-}\mathrm{SH}$ (500 μg/mL), the results were as follows: 0.67% NaCl after 2 h of incubation and 0.68% after 24 h of incubation. For ${\mathrm{SiO}}_{2}\text{-}\mathrm{COOH}$ (500 μg/mL), 50% hemolysis of RBCs was noticed at a NaCl concentration of 0.70% after 2 h of incubation and 0.66% NaCl after 24 h of incubation.

The results revealed that ${\mathrm{SiO}}_{2}$ nanoparticles at concentrations of 200 and 500 μg/mL after an incubation period of 2 h with RBCs, and ${\mathrm{SiO}}_{2}\text{-}\mathrm{COOH}$ at a concentration of 200 μg/mL after an incubation period of 2 h with RBCs, are responsible for an increase in osmotic resistance and, consequently, a decrease in the level of hemolysis (a lower level of hemolysis in relation to the control sample, which means that nanoparticles are safe for red blood cells). In opposition to the observation mentioned above, the results obtained for the rest of the nanoparticles indicated that the resistance of erythrocytes decreased, which suggests a shift in the hemolytic curve towards higher NaCl concentrations (Figure 6) (results for the concentration 200 μg/mL are not shown).

### 2.6. Microscopic Studies of Erythrocyte Shapes

The effects of silica nanoparticles on the shape of erythrocytes were studied using a microscope. Photographs taken during particular stages of the experiment illustrated the changes in red blood cells caused by the presence of the compounds analyzed herein (in particular, changes in the shape of erythrocytes). A selection of photographs, taken using a SEM (scanning electron microscope), are presented in Figure 7. As observed, whereas in relation to the control sample a large number of cells were of regular shapes (discocytes), in the presence of ${\mathrm{SiO}}_{2}$ nanoparticles at concentrations of 200 μg/mL and 500 μg/mL (after 2 h of incubation), echinocytes were a predominant group. Moreover, discocytes also formed. When analyzing ${\mathrm{SiO}}_{2}\text{-}\mathrm{SH}$ and ${\mathrm{SiO}}_{2}\text{-}\mathrm{COOH}$ at both concentrations used, echinocytes, discocytes, and lacrimocytes were noticed, whereas lacrimocytes constituted only a minor part of the erythrocytes. Furthermore, it was also noticed that, at a concentration of 500 μg/mL, erythrocytes had a tendency to aggregate and form clumps of cells. Apart from this, nanoparticles accumulated specifically on the surface of the cell membrane. Both selected concentrations in all mesoporous silica nanoparticles used in the experiment induced the shape of echinocytes and stomatocytes. Bearing in mind the toxicity of silica nanoparticles towards RBCs, we decided not to perform a microscopic examination for an incubation time of 24 h. Furthermore, we observed that in the case of ${\mathrm{SiO}}_{2}\text{-}\mathrm{SH}$ nanoparticles, the number of echinocytes and stomatocytes was definitely higher than in the case of ${\mathrm{SiO}}_{2}\text{-}\mathrm{COOH}$ at both concentrations. Furthermore, when analyzing ${\mathrm{SiO}}_{2}\text{-}\mathrm{COOH}$, the number of RBCs in the shape of lacrimocytes was higher than in ${\mathrm{SiO}}_{2}\text{-}\mathrm{SH}$ (Table 2). The shape is correlated with the concentration of the nanoparticles in the outer layer of the lipid bilayer, which leads to an increase in size in relation to the inner layer and causes changes in the curvature of the bilayer. We believe this is the main cause, which, in turn, leads to the deformation of the membrane and other irregularities.

### 2.7. Influence of SiO_2_ Nanoparticles on the Physical Properties of the Membrane

The results obtained in reference to the level of fluorescence anisotropy using a DPH probe, which reflected the impact of silicon dioxide nanoparticles on RBCs’ membranes and liposomes, led to the conclusion that there is an increase in fluorescence anisotropy in proportion to the increase in the nanoparticle concentration in suspension (Figure 8). Changes in membrane fluidity can have effects on the cell’s functional properties that might be correlated with the nanomaterial’s (or drug’s) mechanism of action. In this context, membrane fluidity studies were performed by anisotropy using the fluorescent probes DPH and TMA-DPH (commonly used as probes of membrane fluidity). The ability of silica nanoparticles to disturb the membrane structure in different regions can be assessed since these two probes are used in order to report the microfluidity of those sites. Modifications in membrane fluidity can be detected by changes in anisotropy, which reflects perturbations in the probe’s rotational movement caused by changes in the stiffness of its surrounding matrix. DPH (1,6-diphenyl-1,3,5-hexatriene) is incorporated in the hydrophobic regions of the lipid bilayer. TMA-DPH (1-(4-trimethylammonium-phenyl)-6-phenyl-1,3,5-hexatriene p-toluene sulfonate) is anchored in the polar head group region of phospholipids (incorporating the fourth carbon atom in the transient region between the hydrophobic and hydrophilic sections of the bilayer). The Laurdan probes (6-dodecanoyl-2-dimethylaminonaphthalene) were located in the hydrophilic regions of the bilayer. Such differentiated incorporation of the probes gives insight into the structural changes caused by nanomaterials [30,31]. Referring to the control, which on average equaled A = 0.25, an increase in A = 0.37 for cell membranes and A = 0.33 for liposomes at a nanoparticle concentration of 1000 μg/mL was observed. These changes were significant and suggested that there was a decrease in the fluidity of the hydrophobic regions of the membrane, as well as an increase in the ordering of the packing structures present in an erythrocyte membrane, and liposomes at the hydrophobic level. An increasing tendency was also observed in the case of ${\mathrm{SiO}}_{2}\text{-}\mathrm{COOH}$ nanoparticles. In comparison to the control, for which the level of fluorescence anisotropy was, on average, greater than A = 0.25 at the highest concentration of 1000 μg/mL for propylcarboxylic-acid-functionalized silica nanoparticles, the level of anisotropy reached A = 0.31 in interaction with cell membranes and A = 0.29 in interaction with liposomes. Different results were obtained for the impact of propylthiol-functionalized silica on a cell membrane. In line with the increase in the concentration of nanoparticles in suspension, the level of fluorescence anisotropy was at a similar level, whereas with an interaction with liposomes, the level of fluorescence anisotropy slightly decreased. This suggests that there are either no changes in the fluidity of the hydrophobic regions of membranes or a different type of interaction between these nanoparticles and a cell membrane. It is well-known that a thiol group can interact with proteins on the surface of a membrane, and this does not have any significant impact on the fluidity of the hydrocarbon chains of lipids. Such an impact may also suggest that there are no changes in fluorescence anisotropy in the case of RBCs, whereas a slight decrease leads to an increase in the fluidity of a membrane at the level of the hydrocarbon chains of liposomes.

The interaction between the tested nanoparticles and a cell membrane was also studied using a TMA-DPH fluorescence marker. The level of fluorescence anisotropy increased in line with an increase in the concentration of the unmodified ${\mathrm{SiO}}_{2}$ particles, in relation to RBCs’ membranes and liposomes (Figure 9). On the basis of the results obtained, it may be stated that this increase was more significant in the case of liposomes, in which the level of anisotropy (in reference to A = 0.23 for the control) increased to A = 0.34 at the highest concentration (1000 μg/mL). The studies analyzed here showed that while unmodified ${\mathrm{SiO}}_{2}$ nanoparticles have an impact on RBCs’ membranes, liposomal membranes lead to their rigidification and better organization in the hydrophobic regions of a bilayer at the level 4 carbon in a carbohydrate chain. Similarly to the DPH probe, ${\mathrm{SiO}}_{2}\text{-}\mathrm{COOH}$ nanoparticles also have an impact on a liposomal membrane. This was proven by an increase in fluorescence anisotropy; however, it was not so high or significant. In the case of nanoparticles, neither an increase proportional to the concentration of nanoparticles, nor a decrease in fluorescence anisotropy, was observed. This proves that ${\mathrm{SiO}}_{2}\text{-}\mathrm{SH}$ are not responsible for changes in the fluidity of cell membranes and liposomes. It can be concluded that ${\mathrm{SiO}}_{2}\text{-}\mathrm{SH}$ particles have no impact on the lipid bilayer at the level 4 carbon in a carbohydrate chain of lipids; therefore, they do not penetrate the hydrophobic layers of a lipid bilayer.

The general polarization of the Laurdan probe is a parameter that informs us about the ordering of polar heads in a lipid bilayer. The results showed that ${\mathrm{SiO}}_{2}$ nanoparticles are not responsible for changes at the level of hydrophilic heads in a lipid particle in an RBC membrane, or in a liposome membrane. This shows that the GP value does not change in line with an increase in the concentration of the suspension of the tested nanoparticles up to 1000 μg/mL (Figure 10).

Both functionalized nanoparticles, ${\mathrm{SiO}}_{2}\text{-}\mathrm{SH}$ and ${\mathrm{SiO}}_{2}\text{-}\mathrm{COOH}$, are responsible for a significant increase in disorder in the hydrophilic heads. Furthermore, the value of the general polarization (in the case of the Laurdan probe) decreased in proportion to an increase in concentration, and this change was most visible in the case of the interaction between ${\mathrm{SiO}}_{2}\text{-}\mathrm{SH}$ nanoparticles and liposomes, in which, in relation to the main probe, which equaled GP = 0.32, this value decreased to GP = 0.16 at a concentration of 1000 μg/mL. Both a significant increase in the number of disordered liposomes and liquefaction of the outer layers of the lipid bilayer were also observed when using ${\mathrm{SiO}}_{2}\text{-}\mathrm{COOH}$ nanoparticles, which caused a significant decrease in the GP value (compared to the control, this value decreased from 0.32 to 0.19 at the highest concentration of nanoparticles); as for erythrocyte cell membranes, the GP value decreased from 0.24 to 0.12 at the same concentration, namely, 1000 μg/mL.

## 3. Discussion

One of the most promising nanomaterials is mesoporous silicon dioxide particles, which, due to their porous structure and the ability to be functionalized on their surface, are useful in medicine [4]. They are characterized by both high biocompatibility and a low cost of production. According to the literature, they do not accumulate in a body and are easily expelled during urination [5], so their potential toxicity (an issue that is extremely controversial) is connected with the size of the particles, their types, and the concentration used, all of which have an impact on their effectiveness. Therefore, silicon dioxide nanoparticles are a subject of scientific interest, with the intention being to use them in the future in biomedical imagining, photodynamic and photothermal therapies, and, first and foremost, targeted drug delivery [32]. Due to their high potential to be used as drug carriers (e.g., when injected into blood circulation), there are numerous studies focused on the impact of mesoporous silica nanoparticles on morphemic blood components, including erythrocytes and cell membranes. The results obtained have proved that hemolytic toxicity depends on the form of silica nanoparticles’ modification, their concentration and size, and the time of exposure. Silica nanoparticles, which were not modified on their surface, reached the highest level of hemolysis (97.09%) after 24 h at a concentration of 1000 μg/mL. Moreover, after 2 h and the same concentration, the level of their hemolysis was 21.39%. A study by Chen et al. [33] showed that ${\mathrm{SiO}}_{2}$ nanoparticles of 200 nm in a concentration of 1000 μg/mL showed hemolysis at a level of 3.5% (after 3 h of exposure). Similarly, in a concentration of 8000 μg/mL, they showed hemolysis at a level of 53%. Furthermore, an important impact on their hemolytic toxicity was exerted by both factors, namely modifications of the surface and the porosity of nanoparticles. The results obtained by Ferenc et al. [34] showed that adding mesoporous nanoparticles of aminopropyl (SBA-${\mathrm{NH}}_{2}$) mercaptopropyl (SBA-SH), and ethylcarboxyl (SBA-COOH) groups had an impact on the level of hemolysis. It was proven that, after 2 h of SBA-${\mathrm{NH}}_{2}$ incubation, the level of hemolysis was 50%, whereas, for the other nanoparticles modified using -SH and -COOH, the level of hemolysis did not exceed 2.5%. After a longer period of incubation (up to 24 h), the level of hemolysis was 6% (SBA-SH) and 3.4% (SBA-COOH). Our research showed that ${\mathrm{SiO}}_{2}\text{-}\mathrm{COOH}$ nanoparticles were not harmful to blood cells, as the level of hemolysis was only 4.53% after 2 h of incubation and in a concentration of 100 μg/mL, and, in a concentration of 1000 μg/mL, this parameter was 6.04%. After 24 h of exposition, the level of hemolysis was 4.41% in a concentration of 100 μg/mL, and 6.88% in a concentration of 1000 μg/mL. When functionalized using a -COOH group, nanoparticles seem to be an extremely promising material due to their low level of hemolysis and lower cytotoxicity. We suggest that this is the consequence of the ${\mathrm{SiO}}_{2}$ nanoparticles’ surface modification, which leads to a reduction in the number of exposed silanols; the impact of this is harmful and leads to hemolysis [34]. The evaluation of changes in the shape of erythrocytes was conducted using microscopic techniques. The presence of discocytes, echinocytes, stomacytes, and lacrimocytes was reported. In our study, after using ${\mathrm{SiO}}_{2}\text{-}\mathrm{COOH}$ nanoparticles on erythrocytes, a small number of lacrimocytes was also detected (in both concentrations, namely 200 μg/mL and 500 μg/mL) (Figure 7). The number of changes in the shapes of erythrocytes (expressed proportionally) is shown in Table 2. The formation of these pathologically changed red cells promotes the formation of clots [35]. A profound study of the interactions between nanomaterials and blood components is extremely important due to the fact that the circulatory system hosts the first contact between nanoparticles and blood cells. The study of their toxicity towards erythrocytes is fundamental as their injection can cause serious side effects, including hemolysis or blood clots. It is well-known that nanoparticles can interact with a positively charged erythrocyte membrane through negatively charged silanol groups, which may lead to hemolysis [17]. It was also proven that nanoparticles, administered orally, cause lower immunotoxicity than those administered intravenously, intraperitoneally, and subcutaneously, and they reduce the probability of interaction with cells of the immune system [36]. Silica nanoparticles are stable, and their period of circulation, disintegration ability, and ability to release content are all regulated as a way of adding to their surface molecules of polyethylene glycol (PEG) in order to block silanol groups [37]. Silicon dioxide nanoparticles are able to transport drugs in their mesopores and initiate diffusion. As a result, their stability is extremely important for releasing drugs in a controlled manner under a specific external stimulus [38,39,40]. Lin and Haynes, while testing the hemolytic toxicity of MSN (grain size: 25–225 nm), showed that the larger the nanoparticle diameter, the lower the hemolytic activity [41]. Furthermore, Yu et al. [42] showed that it is not only the concentration of nanoparticles, their modification, and their period of exposure that have an impact on the level of hemolysis; their shape is equally important. Yet another study proved that plasma protein adsorption on the surface of nanoparticles leads to the formation of a stable protein coat, namely a protein corona (PC) [43], which results in preventing hemolysis, which is induced by mesoporous silicon nanoparticles. In other words, certain modifications of a nanoparticle surface can be beneficial to blood cells, leading to either reducing or blocking cytotoxicity (this property is also proven in this paper via modification using ${\mathrm{SiO}}_{2}\text{-}\mathrm{COOH}$). The greatest interest in the group of mesoporous silicon dioxide nanoparticles is excited by these of a small grain diameter of 58 nm and at a concentration of 100 μg/mL) [44]; significantly lower interest is connected with those of a grain diameter of 461 nm at a concentration of 2000 μg/mL [45]. Therefore, there was a series of experiments, on the basis of which mesoporous nanoparticles of 200 nm and a concentration range of 0–1000 μg/mL were selected, and the results obtained by our team proved that the concentration of ${\mathrm{SiO}}_{2}$ nanoparticles in their interaction with erythrocytes should not exceed 500 μg/mL. Fluorometric studies proved that the nanoparticles tested here had a significant impact on cell membranes and liposomes due to the fact that silicon dioxide nanoparticles significantly reduced the liquefaction of membranes at the level of hydrocarbon lipid chains. Wei et al. studied the impact of NPs (nanoparticles) on model membranes formed using DOPC. In their paper, a decrease in liquidity was caused by the nanoparticles, which may be explained by the formation of hydrogen bonds between a silanol group present in NPs, and phosphate groups present in phospholipids [46]. In this paper, a correlation between the concentration of nanoparticles (in particular, functionalized ones) and the tested membranes, both biological and model, was proven. Changes in the liquidity and order of membranes, in line with an increase in concentration, were observed. Both ${\mathrm{SiO}}_{2}$ and ${\mathrm{SiO}}_{2}\text{-}\mathrm{COOH}$ nanoparticles were mainly responsible for a decrease in liquidity of a membrane in its hydrophobic part, in line with an increase in concentration (the highest concentration used was 1000 μg/mL). Shin et al. proved that there is a correlation between a higher concentration of silica used and the higher toxicity of cells, which was caused by an increase in lipid peroxidation in cell membranes after using a concentration of 1000 μg/mL (whereas, at a concentration of 100 μg/mL, changes were not noticed). Furthermore, an increase in membrane lipid oxidation was observed (phospholipids, glycolipids, sterols), which led to a decrease in both the liquefaction and permeability of a membrane [47]. These nanoparticles modified using a thiol group did not cause any changes within the hydrophobic part of a membrane. Both modified and unmodified silicon dioxide particles had an impact on the structure of erythrocytes and lipid membranes. These changes were similar in erythrocyte membranes and lipid membranes, which suggests that the impact of nanoparticles was mainly on the lipids present in the tested structures. One of the functions of cell membranes is to regulate and control the transport of substances inside and outside of a cell. When these mechanisms of transportation are not stable, the cell is prone to damage. Moreover, a decrease in cell membrane fluidity in erythrocytes may cause a disproportion in the composition of the two sides of a membrane and disallow a change in the shape of erythrocytes, which is extremely important in red blood cells due to microcirculation in blood vessels [10]. To conclude, despite numerous beneficial properties and the potentially broad usage of silicon dioxide nanoparticles, it is necessary to carefully study their impact on the human body within a broad range of interactions at both a molecular and cell level. Each new surface functionalization offers a plethora of new possibilities; however, the size, concentration, period of exposure, and particular types of cells on which they have an impact should be taken into consideration.

## 4. Materials and Methods

### 4.1. Silica Nanoparticles

For this research, the following nanoparticles were used: silica nanoparticles, mesoporous (${\mathrm{SiO}}_{2}$)—product no. 748161, propylthiol-functionalized silica nanoparticles, mesoporous (${\mathrm{SiO}}_{2}\text{-}\mathrm{SH}$)—product no. 749362, propylcarboxylic-acid-functionalized silica nanoparticles (${\mathrm{SiO}}_{2}\text{-}\mathrm{COOH}$)—product no. 749699 (this product is a mixture of silica nanoparticles, mesoporous, propylcarboxylic-acid-functionalized, and meso-Tetra(4-carboxyphenyl)porphine-labeled). All nanomaterials were purchased from Sigma-Aldrich (particle size: 200 nm and pore size: 4 nm).

### 4.2. Fourier Transform Infrared Spectroscopy

In order to determine the quality of the functional groups present in the structures of pure silica, functionalized using propylcarboxylic acid and propylthiol, Attenuated Total Reflectance Fourier Transform Infrared Spectroscopy (ATR-FTIR) was used. In order to conduct this study, a VETEX 70 apparatus produced by Bruker (Mannhaim, Germany) was used. This spectrometer guarantees the extremely high quality of the scanned samples, which exceeds 0.5 ${\mathrm{cm}}^{-1}$. The abovementioned analysis was conducted within the wavenumber range 4000–450 ${\mathrm{cm}}^{-1}$, and the results obtained are presented in the form of infrared spectra.

### 4.3. Particles Size Measurement

The study of silicon dioxides’ dispersion properties was conducted using a Zetasizer Nano ZS, produced by Malvern Instruments, Ltd. (Malvern, UK). In this apparatus, a noninvasive backscatter technology is used, which allows us to measure the size of particles within a range of 0.6 to 6000 nm. In order to conduct this study, the tested samples were dispersed in isopropanol using ultrasounds. To obtain the ${\mathrm{SiO}}_{2}$ dispersion, 10 mg of the material were suspended in 25 mL of isopropanol and the system was sonicated for 15 min in an ultrasonic bath. To produce the dispersion, an ultrasonic bath from POLSONIC Palczyński Sp. J. (model Sonic-3), with an ultrasonic power of 160 W and a frequency of 40 kHz, was used. Next, the samples prepared in this way were placed in the apparatus and the measurement was conducted.

### 4.4. Electrokinetic Potential Measurement

The electrokinetic potential of the analyzed samples of silica was also tested using the Zetasizer Nano ZS. In order to determine the zeta potential, in the first step, the material was dispersed in a buffer solution, i.e., DPBS, by running an ultrasonic bath. In the second step, the pH was measured; in every sample, it was 7.33 ± 0.04. After this stage, the samples tested here were placed in a cuvette fixed in the apparatus handle and the measurement of electrokinetic potential was conducted.

### 4.5. Erythrocytes

The studies were conducted using pig red blood cells due to the fact that these cells are the closest to human erythrocytes, and there were no problems accessing this blood. Whenever erythrocytes were obtained from fresh, heparinized pig blood, they were washed in an isotonic phosphate solution of pH 7.4.

#### 4.5.1. Hemolysis of Erythrocytes

The hemolytic assay was prepared following the method described by Pruchnik et al. [48]. Full blood (fresh, heparinized pig blood) was first centrifuged at 2500 rpm for 3 min at 4 °C (with the intention of removing plasma and leukocytes), and after that RBCs were washed three times with a cold phosphate-buffered saline isotonic solution (PBS: 310 mOsm, pH 7.4). The test sample (1 mL) contained an appropriate volume of phosphate-buffered solution, the silica nanoparticle compounds, and erythrocytes at a final hematocrit level of 1.2%. The hemolytic activity of nanoparticles (NPC) was determined for the range of concentration from 0 to 1000 μg/mL, for 2 and 24 h periods of incubation at 37 °C. In the following step, 2 mL of phosphate buffer (pH 7.4) were added and the samples were centrifuged (2500 rpm, 15 min) at room temperature (RT). After that, the supernatant was assayed for the hemoglobin content using a spectrophotometer on the UV–Vis spectrophotometer (Specord 40, Analytik Jena, Jena, Germany) at a wavelength of 540 nm. The concentration of hemoglobin in the supernatant was expressed as a percentage of the hemoglobin concentration in the supernatant of totally hemolyzed cells and was assumed to be a measure of the extent of hemolysis. Samples with total hemolysis (100%) were prepared as a way of adding deionized water (${\mathrm{dH}}_{2}O$) to the negative control samples.

#### 4.5.2. Osmotic Resistance Assay

We performed the osmotic resistance assay with fresh pig blood, prepared in the same manner as during hemolysis. Firstly, a 1.2% red cell suspension containing silica nanoparticles of 200 and 500 μg/mL concentrations was prepared in 0.9% NaCl and left for 2 or 24 h at 37 ºC with continuous gentle stirring. After that period of incubation, the suspension of erythrocytes was centrifuged (15 min at RT) with an intention to remove cells from ${\mathrm{SiO}}_{2}$. From the cell sediment, 100 μL samples of the extract-modified cells were removed and suspended in test tubes that contained NaCl solutions at concentrations of 0.5–0.86% and 0.9% (isotonic). In the solution, the same concentrations were also suspended in unmodified erythrocytes, which constituted a control for osmotic resistance determination. Subsequently, the suspension obtained was stirred and centrifuged under the conditions described above. Afterwards, the percentage of hemolysis was measured using a spectrophotometer at λ = 540 nm. In this way, one can determine the correlation between the percentage of hemolysis and the NaCl concentration in the solution. The test conducted here proved that, whenever a determined sodium chloride concentration is higher than that of control cells, the osmotic resistance of the red blood cells is lower, and vice versa [23,49].

#### 4.5.3. Microscopic Study of Erythrocyte Shapes

The effect of the extract on the shape of RBCs was determined using an optical microscope. The erythrocytes, separated from plasma, were washed three times in saline solution after being suspended in the same solution, but this time containing 200 or 500 μg/mL silica nanoparticles. The hematocrit level of the erythrocytes in the modified solution was determined to be 1.2% and the period of modification was 2 h or 24 h at 37 °C. After that period of modification, erythrocytes were fixed with a 0.25% solution of glutaraldehyde. Subsequently, the shapes of red blood cells were analyzed under an optical microscope (Nikon Eclipse E200, Melville, NY, USA) equipped with a digital camera. The photographs allowed us to count erythrocytes of various shapes, and after that the percentage share of the three basic forms (discocytes, echinocytes, stomatocytes and lacrimocytes) in a population of c. 300 cells was calculated. Individual forms of the erythrocyte cells were ascribed to morphological indices following the Bessis scale [21]. Furthermore, changes in the shapes of RBCs were also studied using a scanning electron microscope (SEM). In order to be successful in our studies, erythrocytes were prepared in a similar way to those studied using an optical microscope, as described above. Next, erythrocytes were fixed with a 4% solution of glutaraldehyde diluted in a 0.2 M phosphate buffer (pH 7.4) over a period of 8 h. Next, the analyzed material was flushed in a 0.2 M phosphate buffer (pH 7.4) over a period of 24 h. Afterwards, the tested samples were put into 2% osmium tetroxide (OsO4) over a period of 2 h and flushed with distilled water over another 30 min. In the last step, samples were deoxidized in a series of alcohols with a broad range of concentrations, namely, 50%, 70%, 80%, 90%, and, three times, 100%; over a period of 15 min for each of these concentrations. Furthermore, samples were placed on a cover glass, sprinkled with hexamethyldisilazane, and dried in the air. At the end of the experiment, the material was fixed to the subject table of a microscope and sprayed using a vacuum sprayer with both spectrally pure carbon and silver (Ag). Finally, samples were analyzed using a scanning electron microscope (SEM) Tesla–300, as shown in Figure 7.

#### 4.5.4. Scanning Electron Microscopy (SEM)

The silicon dioxide nanoparticles were analyzed using a scanning electron microscope (SEM). Samples of silica powder were placed on the carbon tape on the holder and the images were performed using a FE SEM ZEISS ULTRA PLUS (China) device.

#### 4.5.5. Fluorimetric Studies of the Interaction of the Silica Nanoparticles with Biological and Model Membranes

The impact of silica nanoparticles on the fluidity and mobility/hydration of lipids in the RBCs membrane (ghosts) was investigated using fluorimetric methods. The intensity of fluorescence was measured using fluorescent probes: DPH, TMA-DPH, and Laurdan. These three probes were located in different regions of the lipid bilayer, and fluorescence anisotropy indicated membrane fluidity by free rotation of the lipid probe in the lipid bilayer. It is well-known that an increase in fluorescence anisotropy is correlated with a decrease in the mobility of lipid hydrocarbon chains’ order, showing an increase in membrane fluidity. The erythrocyte membranes (ghosts) were isolated by the method of Dodge et al. [50]. The erythrocyte membranes were suspended in isotonic phosphate-buffered saline at a pH of 7.4. The Bradford method was used to determine the content of membranes in the tested samples [51]. The lipids were extracted from the erythrocytes’ (RBCL) membrane according to the method reported by Maddy et al. [52]. Fluorescence intensity was measured for three probes (DPH, TMA-DPH, and Laurdan) at a sample concentration of 1 μM, while the silica nanoparticles concentration was in the range 0–1000 μg/mL. Measurements were made in quartz cuvettes using a CARY Eclipse fluorimeter (Varian, San Diego, CA, USA) with a DBS Peltier module for sample temperature control. Measurements were made in quartz cuvettes using a fluorimeter (Cary Eclipse, Varian) at excitation and emission wavelengths, respectively, for 1,6-diphenyl-1,3,5-hexatriene (DPH) λex 360 nm, λem 425 nm; 1-(4-trimethylammonium-phenyl)-6-phenyl-1,3,5-hexatriene p-toluene sulfonate (TMA- DPH) λex 358 nm, λem 428 nm; and 6-dodecanoyl-2-dimethylaminonaphthalene (Laurdan) λex 360 nm, λem 440 and 490 nm. Fluorescence anisotropy for probes DPH and TMA-DPH and generalized polarization for Laurdan were calculated by the methods described in the study of Bonarska-Kujawa et al. [53,54].

### 4.6. Statistical Analysis

All data are the mean and SD (standard deviation) from at least three independent experiments. Statistical evaluation of differences was made using ANOVA I and Tukey’s post hoc tests, at significance levels of *p* < 0.05 (*).

## 5. Conclusions

Although ${\mathrm{SiO}}_{2}$ and ${\mathrm{SiO}}_{2}\text{-}\mathrm{SH}$ (modified on its surface with thiol groups) both induced the process of hemolysis depending on their concentration and the period of incubation, ${\mathrm{SiO}}_{2}\text{-}\mathrm{COOH}$ did not show these properties. Both types of nanoparticles, namely, ${\mathrm{SiO}}_{2}$ and ${\mathrm{SiO}}_{2}\text{-}\mathrm{SH}$, reduced/modulated the osmotic resistance of erythrocytes, though to a different degree. Whereas ${\mathrm{SiO}}_{2}\text{-}\mathrm{SH}$, used in concentrations of 200 μg/mL and 500 μg/mL after 2 h of incubation with erythrocytes, induced a significant decrease in their resistance, this was in contrast with the changes caused by ${\mathrm{SiO}}_{2}$. ${\mathrm{SiO}}_{2}\text{-}\mathrm{COOH}$, at a concentration of 200 μg/mL and after 24 h of incubation, only caused an insignificant increase in the resistance of erythrocytes. All of the nanoparticles used induced changes in the shape of erythrocytes, mainly causing the formation of echinocytes. Moreover, they had an impact on both changes in fluidity and the ordering of lipids in lipid membranes, in line with an increase in the concentration of suspension. Both ${\mathrm{SiO}}_{2}$ and ${\mathrm{SiO}}_{2}\text{-}\mathrm{COOH}$ nanoparticles caused a reduction in liquefaction, as well as an increase in the ordering of a cell membrane and in the hydrophobic part of a phospholipid bilayer (proven by the results of fluorescence anisotropy for DPH and TMA-DPH probes). ${\mathrm{SiO}}_{2}\text{-}\mathrm{SH}$ did not cause changes in liquefaction and ordering in the hydrophilic part of cell membranes (seen also in regard to ${\mathrm{SiO}}_{2}\text{-}\mathrm{COOH}$), as well as an increase in the lack of order in the lipid heads.

## Figures and Tables

**Figure 1 ijms-24-11760-f001:**
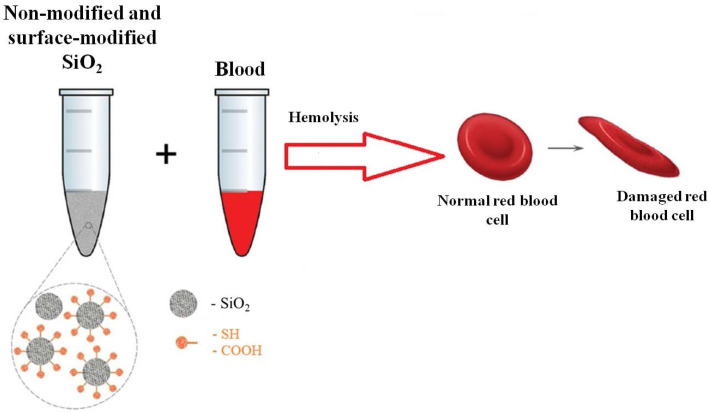
A schematic diagram illustrating the interaction between non-modified and modified silica nanoparticles with red blood cells (RBCs) [17,18].

**Figure 2 ijms-24-11760-f002:**
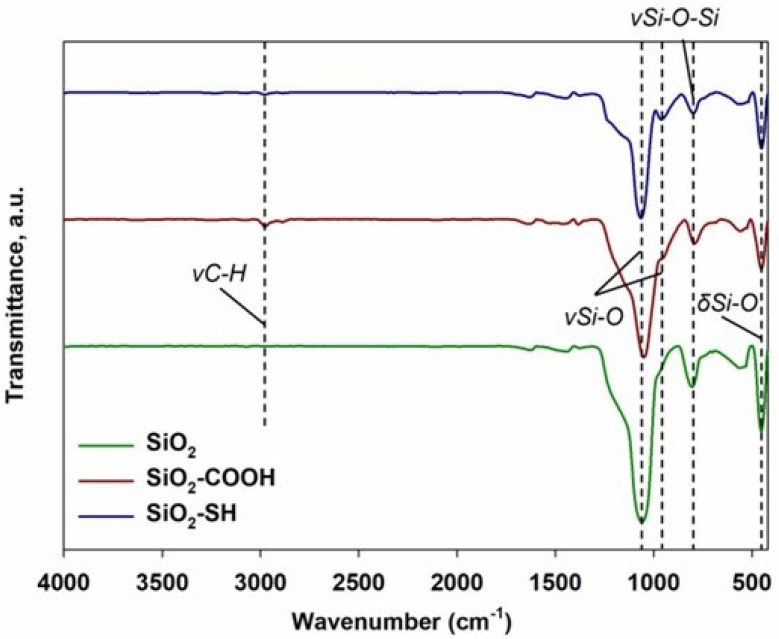
ATR-FTIR spectrum of the samples ${\mathrm{SiO}}_{2}$, ${\mathrm{SiO}}_{2}\text{-}\mathrm{COOH}$ and ${\mathrm{SiO}}_{2}\text{-}\mathrm{SH}$. The data are mean and S.D. from at least 3 independent experiments.

**Figure 3 ijms-24-11760-f003:**
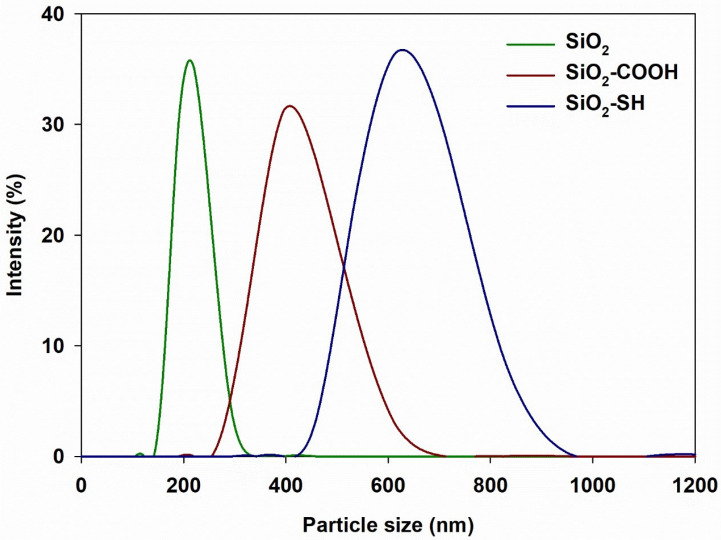
The distributions of size for particles for pure silica, silica functionalized using propylcarboxylic acid and silica functionalized using propylthiol. The data are mean and S.D. from at least 3 independent experiments.

**Figure 4 ijms-24-11760-f004:**
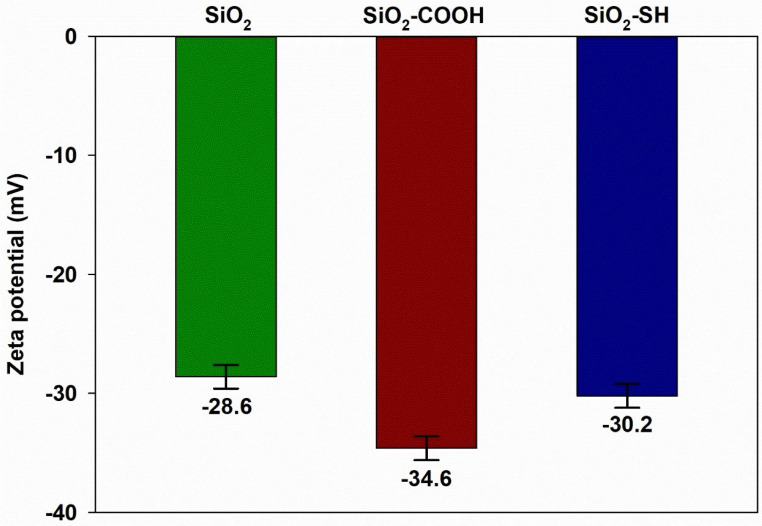
The value of elektrokinetic potential for ${\mathrm{SiO}}_{2}$, ${\mathrm{SiO}}_{2}\text{-}\mathrm{COOH}$ and ${\mathrm{SiO}}_{2}\text{-}\mathrm{SH}$. The data are mean and S.D. from at least 3 independent experiments.

**Figure 5 ijms-24-11760-f005:**
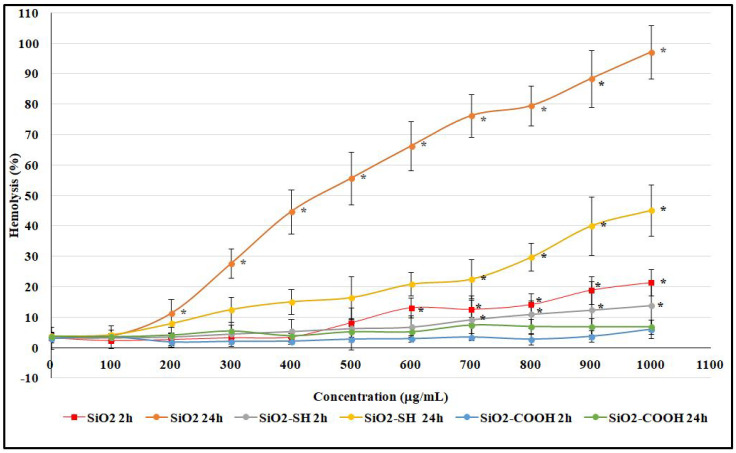
Percentage of hemolysis of erythrocytes The hemolytic toxicity of ${\mathrm{SiO}}_{2}$ nanoparticles concerning red blood cells was analyzed after 2 h and 24 h of incubation. Statistical evaluation of differences was made using the ANOVA I and Tukey’s post hoc test, at significance levels of *p* < 0.05 (*) with respect to control. The data are mean and S.D. from at least 3 independent experiments.

**Figure 6 ijms-24-11760-f006:**
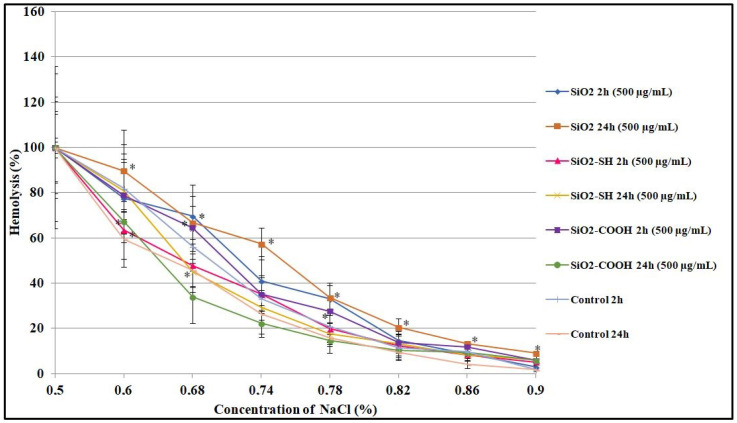
Percentage of hemolysis of erythrocytes. The osmotic resistance of red blood cells modified by ${\mathrm{SiO}}_{2}$ nanoparticles after 2 h and 24 h of incubation. Statistical evaluation of differences was made using the ANOVA I and Tukey’s post hoc test, at significance levels of *p* < 0.05 (*) with respect to control. The data are mean and S.D. from at least 3 independent experiments.

**Figure 7 ijms-24-11760-f007:**
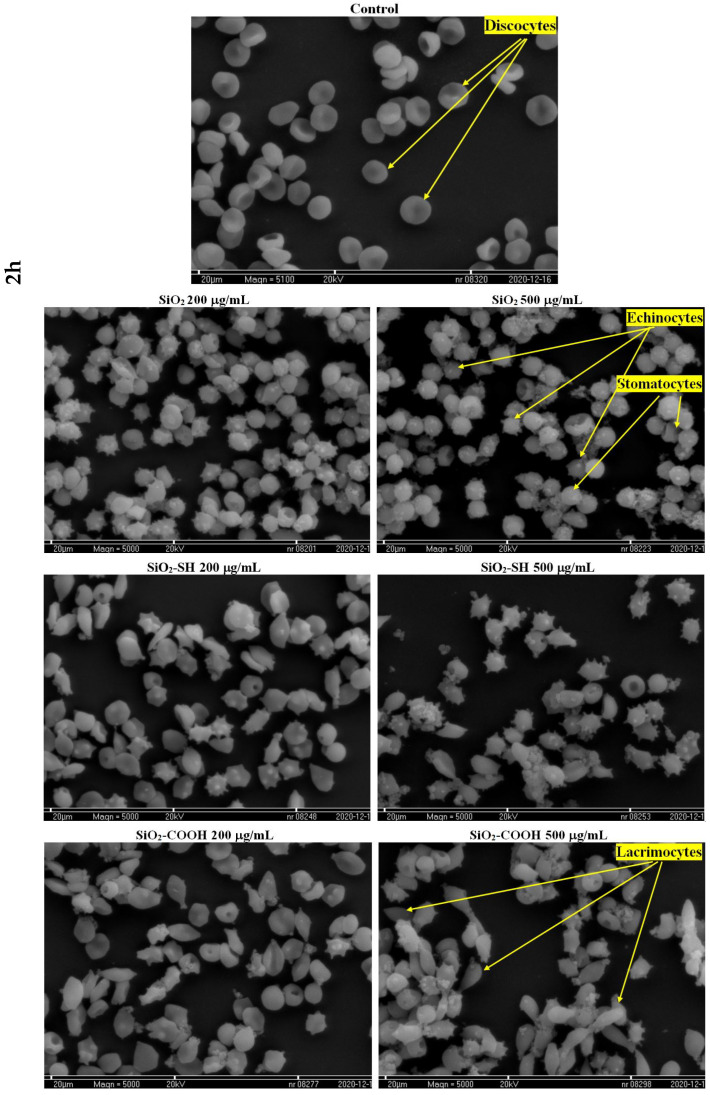
Change of erythrocyte’s shape after treatment red blood cells with different concentration of ${\mathrm{SiO}}_{2}$ and 2 h incubation: control, ${\mathrm{SiO}}_{2}$ 200 μg/mL and 500 μg/mL, ${\mathrm{SiO}}_{2}\text{-}\mathrm{SH}$ 200 μg/mL and 500 μg/mL, ${\mathrm{SiO}}_{2}\text{-}\mathrm{COOH}$ 200 μg/mL and 500 μg/mL. The scale bar corresponds to 20 μm. To enhance visualization of the morphological changes in red blood cells, the designated area of the images (located above) was subjected to zooming. The data are mean and S.D. from at least 3 independent experiments.

**Figure 8 ijms-24-11760-f008:**
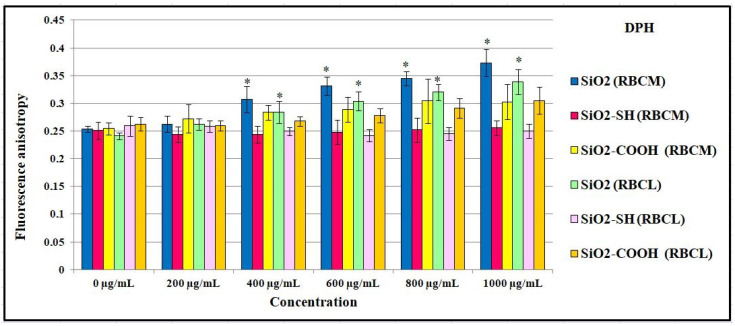
Values of fluorescence anisotropy of DPH probe for erythrocyte membrane and liposomes incubated with silica nanoparticles in the concentrations range 0–1000 μg/mL, at 37 °C, after 2 h. Statistical evaluation of differences was made using the ANOVA I and Tukey’s post hoc test, at significance levels of *p* < 0.05 (*) with respect to control membranes. The data are mean and S.D. from at least 3 independent experiments.

**Figure 9 ijms-24-11760-f009:**
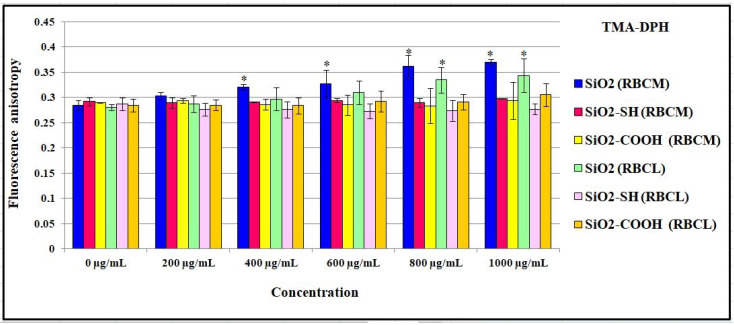
Values of fluorescence anisotropy of TMA-DPH probe for erythrocyte membrane and liposomes incubated with silica nanoparticles in the concentrations range 0–1000 μg/mL, at 37 °C, after 2 h. Statistical evaluation of differences was made using the ANOVA I and Tukey’s post hoc test, at significance levels of *p* < 0.05 (*) with respect to control membranes. The data are mean and S.D. from at least 3 independent experiments.

**Figure 10 ijms-24-11760-f010:**
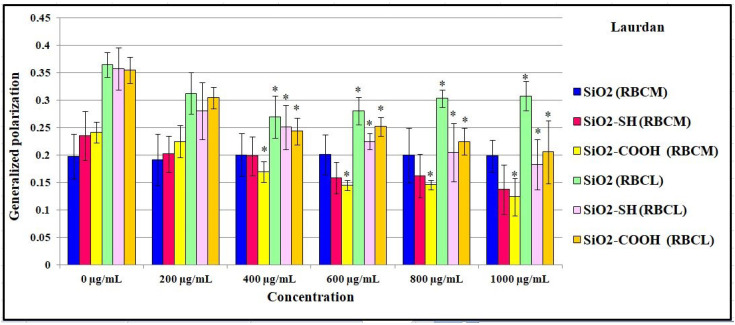
Values of fluorescence anisotropy of Laurdan probe for erythrocyte membrane and liposomes incubated with silica nanoparticles in the concentrations range 0–1000 μg/mL, at 37 °C, after 2 h. Statistical evaluation of differences was made using the ANOVA I and Tukey’s post hoc test, at significance levels of *p* < 0.05 (*) with respect to control membranes. The data are mean and S.D. from at least 3 independent experiments.

**Table 1 ijms-24-11760-t001:** The level of osmotic resistance in relation to the concentration of psychological saline (${\mathrm{IC}}_{50}$), which is responsible for 50% hemolysis of erythrocytes that were modified by ${\mathrm{SiO}}_{2}$ nanoparticles. The data are mean and S.D. from at least 3 independent experiments.

${\mathrm{IC}}_{50}$ (μg/mL)		
**Time incubation**	**2 h**	**24 h**
**Control**	0.69% NaCl	0.66% NaCl
**${\mathrm{SiO}}_{2}$ 200 μg/mL**	0.69% NaCl	0.68% NaCl
**${\mathrm{SiO}}_{2}$ 500 μg/mL**	0.71% NaCl	0.73% NaCl
**${\mathrm{SiO}}_{2}\text{-}\mathrm{SH}$ 200 μg/mL**	0.66% NaCl	0.66% NaCl
**${\mathrm{SiO}}_{2}\text{-}\mathrm{SH}$ 500 μg/mL**	0.67% NaCl	0.68% NaCl
**${\mathrm{SiO}}_{2}\text{-}\mathrm{COOH}$ 200 μg/mL**	0.69% NaCl	0.64% NaCl
**${\mathrm{SiO}}_{2}\text{-}\mathrm{COOH}$ 500 μg/mL**	0.70% NaCl	0.66% NaCl

**Table 2 ijms-24-11760-t002:** The microscopic studies of erythrocytes shapes. The results prove that mesoporous silica nanoparticles are responsible for changes in the shape of erythrocytes after 2 h of exposure. The data are mean and S.D. from at least 3 independent experiments.

Change of Erythrocyte’s Shape				
**Time Incubation (2 h)**	**Discocytes**	**Echinocytes**	**Stomatocytes**	**Lacrimocytes**
**Control **	95.18 ± 0.011%	3.09 ± 0.013%	1.23 ± 0.017%	-
**${\mathrm{SiO}}_{2}$ 200 μg/mL**	8.13 ± 0.015%	74.21 ± 0.015%	17.15 ± 0.014%	-
**${\mathrm{SiO}}_{2}$ 500 μg/mL**	5.11 ± 0.014%	76.14 ± 0.011%	19.13 ± 0.016%	-
**${\mathrm{SiO}}_{2}\text{-}\mathrm{SH}$ 200 μg/mL**	7.18 ± 0.020%	65.23 ± 0.017%	26.11 ± 0.011%	2.24 ± 0.013%
**${\mathrm{SiO}}_{2}\text{-}\mathrm{SH}$ 500 μg/mL**	5.22 ± 0.011%	60.33 ± 0.014%	28.31 ± 0.013%	5.14 ± 0.012%
**${\mathrm{SiO}}_{2}\text{-}\mathrm{COOH}$ 200 μg/mL**	12.12 ± 0.011%	52.14 ± 0.015%	23.11 ± 0.013%	13.32 ± 0.015%
**${\mathrm{SiO}}_{2}\text{-}\mathrm{COOH}$ 500 μg/mL**	2.32 ± 0.011%	55.13 ± 0.015%	25.21 ± 0.015%	18.21 ± 0.0143%

## Data Availability

Data sharing is not applicable to this article.

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
