# Peer review of "Effect of Non-Modified as Well as Surface-Modified SiO2 Nanoparticles on Red Blood Cells, Biological and Model Membranes"

_ijms, 2023, doi:10.3390/ijms241411760_

Round 1

Reviewer 1 Report

The authors have done exciting work. They have well designed and executed the experiments; I recommend this article for publication after revising the manuscript based on the following queries.

Can the authors provide the TEM images of all the silica and functionalized silica nanoparticles for understanding the interaction with erythrocytes?

Apart from functionalization, I think the nature of the porosity of nanoparticles also plays a vital role in the interaction with RBCs. Can the author discuss this in detail?

A schematic diagram of the interaction of silica and different functionalized silica nanoparticles with RBCs can be made with the mechanism. 

Author Response

Dear Editor and Reviewers of Molecules MDPI journal,

The authors are very grateful to the Reviewers for their comments and suggestions. We hope that after correction the whole text will meet the Reviewers expectations and it will be fully accepted.

With due respect

Katarzyna Solarska-Ściuk

Reviewer 2 Report

The article "Effect of non-modified as well as surface-modified SiO2 nanoparticles on red blood cells, biological and model membranes" is an interesting study on a very relevant topic. In general, the article contains interesting and high-quality material. However, there are fundamental remarks that require additional experiments and revision of the text:

A nanoparticle is defined as a particle of matter that is between 1 and 100 nanometres (nm) in diameter (https://doi.org/10.1351%2FPAC-REC-10-12-04). According to Figure 2, the authors do not use such particles (the minimum size is 142 nm), so the entire ideology of the article needs to be revised. The term "nanoparticles" in the title and text of the article should be abandoned.

The introduction to the article needs to be finalized and a better analysis of the literature data on the effect of nanoparticles and microparticles on red blood cells and biological membranes.

The conclusion about the presence of water on the surface of silicon dioxide particles (lines 104–106) is rather disputable, since there are no characteristic vibration bands in the region of 3400 cm–1.

In the presented article, there are very few results on the study of the structure of the obtained materials by the method of scanning electron microscopy. What is the uniformity of the particle structure? What is the pore size on the surface? These are serious questions that affect their biological characteristics.

In Figure 6, the bar-mark is incomprehensible and it is difficult to determine the sizes of objects.

The connection between the functionalization of the particles used and their biological properties is unclear. How exactly do functional groups affect red blood cells and membranes? Does the formation of new chemical bonds occur during their interaction? Or is the difference in the effect of the particles simply due to their different sizes?

Author Response

(The authors gave the same response as above.)

Reviewer 3 Report

In this manuscript the authors assessed the cytotoxic effects of three different commercial silica nanoparticles on red blood cells. The nanoparticles investigated included unmodified SiO2, as well as those modified with thiol and carboxylic acid groups. The authors conducted a brief characterization of the nanoparticles before subjecting them to cytotoxicity testing. Furthermore, the authors employed scanning electron microscopy (SEM) to examine the cellular morphology following exposure to the nanoparticles. To investigate potential interactions, the authors monitored changes in membrane fluidity by studying the fluorescently labeled membranes in the presence of the nanoparticles. The manuscript may be of interesting for publication by taking some arguments and questions into consideration.

1) To elucidate the disparities in chemical functional groups among three silica nanoparticles, it is crucial to present distinct data for each nanoparticle in a clear and discernible manner for readers. Emphasizing the variances in functional groups, such as C-H, C=O, C-O, O-H, C-S, and S-H, should be the main focus of the discussions. The IR spectra provided by the authors were too alike, making it challenging to perceive the dissimilarities.

2) The authors asserted that the nanoparticles' sizes exhibited significant variations as a result of aggregation in the solution. Size holds substantial importance when assessing the cytotoxicity of a nanoparticle. Even when possessing the same particle size, aggregates with differing sizes will present distinct contacting surfaces, solubilities, morphologies, and other characteristics. Therefore, functional group alone is not the only factor influencing cytotoxicity or hemolysis in this study. To resolve this argument, it is essential to provide appropriate comparisons or explanations that address these additional factors.

3) The experimental methodology for particle size measurement lacks specific details. The concentration of the samples, duration and intensity of sonication are critical factors that can influence the size of the particle aggregates.

4) The authors conducted an evaluation of the zeta potential of the nanoparticles; however, it appears that no additional investigations were performed to correlate the stability and cytotoxicity of the nanoparticles. It raises the question of why stability was assessed in the first place. Understanding the stability of nanoparticles is crucial as it directly influences their potential cytotoxic effects. How exactly does the stability of nanoparticles impact their cytotoxicity?

5) Considering that the evaluation of hemolysis and osmotic resistance is aimed at demonstrating the cytotoxicity of the various nanoparticles. It appears that there is an inconsistency (from toxicity perspective) in the experimental results presented in sections 2.4 and 2.5. To facilitate better understanding, a clear and appropriate explanation for this discrepancy should be provided.

6) The English in the paper requires thorough proofreading and refinement. Numerous grammar and language issues have arisen throughout the text. Additionally, certain words and expressions lack the required scientific and professional tone. Furthermore, typographical errors can be found in line 82-83 and line 179. It is necessary to address these concerns to ensure the paper's clarity and professionalism.

The English in the paper requires thorough proofreading and refinement. Numerous grammar and language issues have arisen throughout the text. Additionally, certain words and expressions lack the required scientific and professional tone. Furthermore, typographical errors can be found in line 82-83 and line 179. It is necessary to address these concerns to ensure the paper's clarity and professionalism.

Author Response

(The authors gave the same response as above.)

Round 2

Reviewer 3 Report

Authors have answered and resolved all the issues in the original manuscript. 

English is fine to read.